# Reno-protective effect of IL-34 inhibition on cisplatin-induced nephrotoxicity in mice

**Yukihiro Wada** [1] *, **Masayuki Iyoda**[1,2], **Kei Matsumoto**[3], **Taihei Suzuki**[1],
**Shohei Tachibana**[1], **Nobuhiro Kanazawa**[1], **Hirokazu Honda**[1]

**1** Division of Nephrology, Department of Medicine, Showa University School of Medicine, Tokyo, Japan,
**2** Department of Microbiology and Immunology, Showa University School of Medicine, Tokyo, Japan,
**3** Division of Nephrology, Department of Medicine, Showa University Koto Toyosu Hospital, Tokyo, Japan

* yukihiro@med.showa-u.ac.jp

pone.0245340

Biotechnology, ITALY

**Data Availability Statement:** All relevant data are
within the manuscript and its Supporting
Information files.

## Abstract

### Introduction

Interleukin-34 (IL-34) shares a receptor (cFMS) with colony stimulating factor-1 (CSF-1),
and these two ligands mediate macrophage proliferation. However, in contrast to CSF-1,
the influence of IL-34 on tubular epithelial cells (TECs) injury remains unclear. We investi-
gated the physiological effects of IL-34 on TEC damage caused by cisplatin nephrotoxicity
(CP-N).

### Methods

Mice were administered anti-mouse IL-34 antibody (anti-IL-34 Ab; 400 ng/kg) or vehicle
from 1 day before and up to 2 days after CP-N induction. *In vitro*, mouse renal proximal
TECs (MRPTEpiC) were cultured to analyze the inhibitory effects of IL-34 on CP-induced
TEC apoptosis.

### Results

Compared to vehicle treatment, anti-IL-34 Ab treatment significantly suppressed the intra-
renal expression of IL-34 and its two receptors, cFMS and PTP-ζ, and significantly improved
renal function, ameliorated tubulointerstitial injury, suppressed macrophage infiltration, and
reduced apoptotic cell numbers in CP-N mice. It also significantly reduced the renal tran-
script levels of Kim-1, MIP-1/CCL3, TNF-α, and Bax in CP-N mice. Furthermore, anti-IL-34
Ab-treated CP-N mice showed less renal infiltration of F4/80⁺TNF-α⁺ cells. *In vitro*, stimula-
tion with CP induced the expression of IL-34 and its two receptors in MRPTEpiC. Anti-IL-34
Ab treatment significantly suppressed CP-induced Bax expression with the degradation of
ERK1/2 phosphorylation in damaged MRPTEpiC.

### Conclusions

IL-34 secreted from damaged TECs appeared to be involved in the progression of CP-N.
Inhibition of IL-34 with neutralizing antibody directly prevented CP-induced TEC apoptosis
by inhibiting the phosphorylation of ERK 1/2. Blocking of IL-34 appears to suppress the

**Funding:** This study was supported by the Japan Society for the Promotion of Science KAKENHI (Grant number 19K08734).

proliferation of cytotoxic macrophages, which indirectly attenuates CP-N. Thus, IL-34 represents a potential therapeutic target for TEC injury, and the inhibition of IL-34 might have a reno-protective effect.

## Introduction

Interleukin (IL)-34 is a homodimeric cytokine identified in 2008 as a novel ligand for colony-stimulating factor 1 (CSF-1) receptor (CSF-1R; also known as cFMS and CD115) [1–3]. IL-34 and CSF-1 bind to a common receptor, cFMS, and both ligands promote macrophage (Mø) proliferation, survival, and differentiation [2]. Although IL-34 and CSF-1 broadly share similar physiological properties, these two Mø mediators display different expression patterns [4, 5]. Additionally, the affinity of IL-34 for cFMS is reported to be seven-fold stronger than that of CSF-1 [1, 3]. Furthermore, the protein-tyrosine phosphatase ζ receptor (PTP-ζ, PTPRZ1) was discovered as a second receptor of IL-34 in brain [6], and IL-34 signaling through PTP-ζ is thought to be involved in protein tyrosine phosphorylation [6]. The dissimilarities among these two ligands indicate that IL-34 has distinct biological functions from CSF-1.

Similar to CSF-1, the expression of IL-34 is abundant principally in bone marrow (BM) and spleen, but it can be seen in several other organs, including kidney [1, 7]. In addition, cFMS is expressed on promonocytes, monocytes, Møs, dendritic cells, and some epithelial cells [8–12]. PTP-ζ is primarily expressed on neural progenitors, glia, and glioblastoma, but it can also be found in B cells, T cells, monocytes, and epithelial cells [6, 13–15]. In kidney diseases, recent reports using mouse experimental models have revealed that not only CSF-1, but also IL-34 was expressed by damaged tubular epithelial cells (TECs) in ischemia/reperfusion (I/R) injury or lupus nephritis (LN) [13, 15]. Also, the expression of its two receptors, cFMS and PTP-ζ, was up-regulated on damaged mice TECs [13, 15]. Similarly, the tubular cell expression of IL-34, CSF-1, and its two receptors was increased following human kidney graft I/R [13]. Damaged TECs in patients with LN showed the up-regulation of both IL-34 and CSF-1, as well as the up-regulation of cFMS and PTP-ζ [15]. Moreover, a significant positive correlation between elevated serum IL-34 levels and renal insufficiency was shown in patients with chronic kidney disease [16]. Taken together, the intra-renal expression of IL-34 and its receptors could be associated with kidney diseases.

Generally, Møs regulate the inflammatory response [17]. Activated Møs are conceptually divided into two major polarization states: classical type 1 (M1) and alternative type 2 (M2) [18, 19]. Simplistically, M1 Møs are pro-inflammatory or cytotoxic Møs since they secrete pro-inflammatory cytokines, such as tumor necrosis factor-α (TNF-α); in contrast, M2 Møs are anti-inflammatory or cyto-protective Møs since they secrete anti-inflammatory cytokines, such as IL-10 [17–20]. Regarding the contribution of Møs to kidney diseases, a recent review mentioned that aberrantly proliferative M1 Møs aggravates acute kidney injury (AKI), and the inadequate removal of M2 Møs causes renal fibrosis [20]. Additionally, Mø mediators are also involved in the development of kidney diseases. Previous studies using a mouse I/R model revealed that CSF-1 exerted a direct protective effect on TECs, and an indirect protective effect via the promotion of M2 Møs [11, 21, 22]. In contrast, TEC-derived IL-34 was identified as a promoter of TEC destruction in mouse I/R-induced AKI [13] and LN [15]. However, unlike CSF-1, the direct autocrine effect of IL-34 on TECs, and the influence of IL-34 on Mø polarization remain to be undetermined in such experimental models.

Rodent cisplatin-nephrotoxicity (CP-N) is widely accepted as a representative kidney disease model [23], and it has been recognized as a simple and reproducible model with high

clinical relevance [24, 25]. The primary pathophysiology of CP-N is characterized by proximal tubular injury, and TEC apoptosis caused by inflammation, oxidative stress, or vasoconstriction is central to the progression of CP-N [23, 26, 27]. Furthermore, remarkably, the accumulation of Møs in inflamed kidney tissue was revealed in the course of CP-induced AKI [23, 24, 26–29], which affects TEC apoptosis via the mediation of inflammatory responses. Therefore, it was suggested that the experimental CP-N model could be useful for investigating not only TEC damage, but also intra-renal Mø infiltration.

In the present study, we attempted to elucidate the influence of IL-34 on TEC damage and Mø polarization by focusing on the physiological properties of IL-34 in *in vivo* and *in vitro* models of CP-N; no such study has yet been attempted to date. Herein, we demonstrated that the blocking of IL-34 with neutralizing antibody (nAb) attenuated CP-induced TEC apoptosis by inhibiting IL-34 signaling through its two receptors, and preventing cytotoxic Mø proliferation.

## Methods

### Experimental protocol

The experimental protocol for this study was reviewed and approved by the Animal Care Committee of Showa University in Tokyo (Permit number: 09017). Seven-week-old male C57BL/6 (B6) mice weighing 20 to 23 g were purchased from Sankyo Labo Service Corporation, Inc. (Tokyo, Japan) for use in all of the experiments. The mice were housed in the animal care facility of Showa University under standard conditions (25˚C, 50% relative humidity, 12-hour dark/light cycle) with free access to food and water. In addition, mice used in the present study were weighed daily and food-intake was monitored twice daily during the experiment.

A total of 16 mice were fasted for 8 h, then CP-N was induced by intraperitoneal injection (IP) of CP (Sigma-Aldrich, St. Louis, MO, USA) at a dose of 25 mg/kg on day 0. Groups of animals were given either anti-mouse IL-34 antibody (CP+anti-IL-34 Ab, 400 ng/kg, n = 8; #AF5195, R&D Systems, Minneapolis, MN, USA) or vehicle (CP+V, equal volume of saline, n = 8) daily by IP from day -1 (12 h prior to the CP injection) to day 2. The dose of anti-IL-34 Ab was determined based on the manufacturer's data sheet. Moreover, three age-matched male B6 mice, used as normal controls (NCs), were fasted for 8 h, then injected with saline (equal volume as the CP) by IP, followed by the daily administration of saline (equal volume as the anti-IL-34 Ab) by IP from day -1 to day 2. Animal health and well-being were monitored at the point of 1h after every procedure of IP during the experiment. At day 3 (72 h after the CP injection), each mouse was anesthetized and sacrificed by exsanguination after the cardiac puncture; blood was collected by cardiac puncture and kidneys were collected. Renal tissue was divided; separate portions were snap-frozen in liquid nitrogen or fixed in 2% paraformaldehyde/phosphate-buffered saline for later use. All surgery was performed under anesthesia by pentobarbital (100 mg/kg), and all efforts were made to minimize suffering.

### Creatinine determination

Serum creatinine (s-Cr) levels were measured by a Cr assay kit (#DICT-500, BioAssay System, Hayward, CA, USA) as recommended by the manufacturer.

### Light microscopy

We scored the kidney pathology in periodic acid–Schiff (PAS)–stained paraffin sections, as previously detailed [30]. Briefly, luminal hyaline casts were assessed in 10 fields for each

section. The number of casts was counted under 200× magnification, and the mean number per field was calculated. To evaluate the renal tubular injury, 10 fields from each section were evaluated under 400× magnification. The extent of tubular injury was assessed by counting the percentage of areas with tubular dilatation, tubular cast, and tubular epithelial cell necrosis per field. Scores from 0 to 4 were used (0, none; 1, <10%; 2, 10%– 25%; 3, 25%– 50%; 4, >50% of areas injured), and the results were averaged.

## Immunohistochemistry (IHC)

The antibodies used in this study were as follows: rabbit anti-mouse IL-34 (#orb184448, Biorbyt, St. Louis, MO, USA), rabbit anti-mouse CSF-1 (#ab99178, Abcam, Cambridge, UK), rabbit anti-mouse F4/80 (#ab111101, Abcam), and rabbit anti-mouse caspase-3 (#9664, Cell Signaling, Danvers, MA, USA). The EnVisionTM+Dual Link System HRP, based on a horseradish peroxidase (HRP)-labeled polymer, was purchased from Dako (Glostrup, Denmark). Immunohistochemical staining for IL-34 (1:100 dilution) and CSF-1 (1:500 dilution) was performed as previously described [31, 32]. Similarly, the detailed steps for IHC staining of F4/80 (1:500 dilution) and caspase-3 (1:800 dilution) were described in previous reports [30, 33, 34]. Cells undergoing apoptosis were identified by *in situ* terminal deoxynucleotidyl transferase-mediated dUTP nick end labeling (TUNEL) using the Apop-Tag Plus Peroxidase *In Situ* Apoptosis Detection Kit (Chemicon International Inc., Temecula, CA, USA). The intra-renal positivity of IL-34 or CSF-1 was graded on a four-point scale (0, absent; 1, weak; 2, moderate; 3, strong; and 4, very strong) [15, 35] in 10 randomly selected high-power fields (HPFs) under 400× magnification. The quantification of F4/80, caspase-3, and TUNEL-positive cells in the tubulointerstitium was performed as previously described [15, 30]. The mean score was then calculated as the positivity per HPF or the number of positive cells per HPF.

## Homogenization of kidney tissues

Thirty milligrams of protein in kidney tissues (cortex) were homogenized as previously described [36, 37].

## Enzyme-linked immunosorbent assay (ELISA)

To quantify the IL-34 and CSF-1 levels in the kidneys, sera, and supernatants, an IL-34 ELISA kit (#M3400, R&D Systems) and CSF-1 ELISA kit (#EMCSF1, Thermo Scientific, Frederick, MD, USA) were used according to the manufacturers' instructions.

## Real-time reverse transcriptase polymerase chain reaction (RT-PCR)

The gene expression of mouse IL-34, CSF-1, cFMS, PTP-ζ, monocyte chemoattractant protein-1 (MCP-1), macrophage inflammatory protein-1α (MIP-1α), kidney injury molecule-1 (Kim-1), Bax, Bcl-2, TNF-α, IL-6, IL-1β, IL-10, β-actin, and glyceraldehyde-3-phosphate dehydrogenase (GAPDH) was evaluated using real-time RT-PCR (TaqMan) assays. The procedure for real-time RT-PCR has been described previously [30, 33–35]. All primers used in the real-time RT-PCR analysis were purchased from Applied Biosystem (South San Francisco, CA, USA), and detail information of all the primers were summarized in S1 Table. mRNA expression was normalized using β-actin or GAPDH as an endogenous control to correct for the differences in the amount of total RNA added to each reaction.

## Western blot (WB) analysis

Homogenized kidney and harvested lysates from cultured cells were used for WB analysis. For each sample, 10 μg of protein from kidney tissue or 25 μg of protein from cultured cells were separated by sodium dodecyl sulfate-polyacrylamide gel electrophoresis on a 4% to 20% gradient gel (Invitrogen), and the proteins were transferred to a polyvinylidene difluoride membrane. The detailed procedures were described previously [30, 34, 38]. The primary antibodies for cFMS (B-8 #sc-692, 1:500, Santa Cruz Biotechnology, Dallas, Texas, USA), PTP-ζ (#610179, 1:500, BD Transduction Laboratories, San Jose, CA, USA), Bax (#2772, 1:500, Cell Signaling), phosphorylated (p)-extracellular-signal-regulated kinase 1/2 (ERK1/2; #4370, 1:1000, Cell Signaling), total (t)-ERK1/2 (#4695, 1:1000, Cell Signaling), GAPDH (#2118, 1:1000, Cell Signaling), and α-tubulin (Tub; #2144, 1:1000, Cell Signaling) were used. HRP-conjugated anti-rabbit IgG antibody (#7074, 1:3000, Cell Signaling) or HRP-conjugated anti-mouse IgG antibody (#11317, 1:3000, Santa Cruz Biotechnology) was used as the secondary antibody. Changes in each expression level were normalized by correction to the densitometric intensity of GAPDH or α-Tub for each sample.

## Fluorescence-activated cell sorting (FACS)

We prepared and stained single-cell suspensions from both kidneys as described previously [11, 15, 39]. The antibodies used for FACS were as follow: fluorescein isothiocyanate-conjugated rat anti-mouse CD11b (#557396, Becton Dickinson (BD), San Jose, CA, USA), Alexa Fluor 647-conjugated rat anti-mouse F4/80 (#565853, BD), phycoerythrin-conjugated rat anti-mouse TNF-α (#554419, BD), and phycoerythrin-conjugated rat anti-mouse IL-10 (#554467, BD). To examine intracellular cytokine expression, kidneys were incubated in RPMI 1640 containing GolgiStop™ solution (#554715, BD) for 8 h in 5% $CO_2$ at 37°C. The cells were washed in phosphate-buffered saline, suspended in FACS buffer (#554656, BD) with protein-block (Fcg III/II Receptor, 1:100 dilution, BD, #553141), and incubated with cell surface markers for 30 min on ice. Then, the cells were fixed and permeabilized using Fix/Perm buffer (#554715, BD). After the staining of intracellular cytokines, the cells were washed twice in Fix/Perm buffer, and suspended in FACS buffer to allow the resealing of the permeabilized membranes. We collected $1.0 \times 10^5$ to $5.0 \times 10^5$ total kidney cells using FACSCalibur (BD), and analyzed the data using FlowJo software 9.3 (Tree Star, Palo Alto, CA, USA).

## Cell cultures

Mouse renal proximal TECs (MRPTEpiC) derived from B6 mouse kidney were purchased from ScienCell Research Laboratories (San Diego, CA, USA), and mouse leukemic monocytes derived from a BALB/c mouse (RAW 264) were obtained from the RIKEN Cell Bank (Tsukuba, Japan). According to the manufacturer's data sheet, purchased MRPTEpiC (catalog number: M4100-57) were characterized by immunofluorescence with antibodies specific to cytokeratin-18, -19, and vimentin. The MRPTEpiC are negative for mycoplasma, bacteria, yeast, and fungi. MRPTEpiC are guaranteed to further culture under the condition provided by ScienCell Research Laboratories. Also, quality control of RAW 264, provided by RIKEN Cell Bank, is guaranteed (resource number: RBRC-RCB0535). Authentication of the RAW 264 was confirmed by isozyme analysis (Lot31: LD, NP). The RAW 264 are negative for mycoplasma, bacteria, yeast, and fungi.

MRPTEpiC were initially thawed onto poly-L-lysine (#413, ScienCell)-coated six-well plates, and cultured in Epithelial Cell Medium-animal (EpiCM-a, #4131, ScienCell) for 72 to 96 h until 80% to 90% confluent. Thereafter, the medium was changed to DMEM-F-12 (1:1) supplemented with 5% fetal bovine serum (FBS) and antibiotics during the stimulation or

treatment for experiments. RAW 264 were cultured in DMEM-F-12 (1:1) supplemented with 10% FBS and antibiotics. All cells were cultured in an atmosphere of 5% $CO_2$-95% air at 37˚C in a humidified incubator. Experiments were performed with cells up to the second passage for MRPTEpiC and the fifth passage for RAW 264, as it has been shown that there are no phenotypic changes up to these passage numbers [40–42].

## MTT assay

RAW 264 ($2 \times 10^4$/well) were cultured in 96-well plates, and stimulated with CP, recombinant mouse IL-34 (rIL-34, #577609, BioLegend, San Diego, CA, USA), mouse anti-IL-34 Ab (R&D Systems), and MRPTEpiC supernatants for 12 h. We analyzed the proliferation of the stimulated RAW 264 using the MTT colorimetric assay (Roche, Palo Alto, CA, USA) according to the manufacturer's instructions.

## Lactate dehydrogenase (LDH) assay

MRPTEpiC were cultured in 96-well plates at a density of $1 \times 10^4$ to $2 \times 10^4$/mL. Once they reached 70% to 80% confluence, the cells were incubated with serum-free medium for 6 h, followed by stimulation with CP (2 μg/mL). The MRPTEpiC were then treated with anti-IL-34 Ab (1000 pg/mL, R&D Systems), rIL-34 (500 pg/mL, BioLegend), or vehicle (equal volume of saline). Cell injury was assessed using an LDH assay (#4744926001 Cytotoxicity Detection Kit, Roche, Indianapolis, IN, USA) according to the manufacturer's instructions.

## Statistical analyses

Data are shown as the mean ± standard error of the mean (SEM), and were analyzed by GraphPad Prism software, version 5.0 (GraphPad, La Jolla, CA, USA). The Mann-Whitney U test was applied for comparisons between groups. P values of $< 0.05$ were considered to be statistically significant in all the analyses.

## Results

### Stimulation with CP induced the expression of IL-34 and its two receptors in cultured TECs

To determine the expression of IL-34 and its two receptors, cFMS and PTP-ζ, on damaged TECs, cultured MRPTEpiC were stimulated with CP (2 μg/mL) for 0 to 24 h, followed by the harvesting of the cells and the supernatants (Fig 1A). Similar to the stimulation with TNF-α as a positive control, CP stimulation appeared to induce MRPTEpiC injury over time, which was confirmed by the persistent significant elevation of Kim-1 transcripts in MRPTEpiC until 24 h after CP stimulation (6 h: $P < 0.05$ vs. 0 h; 12 h: $P < 0.05$ vs. 0 h; 24 h: $P < 0.05$ vs. 0 h; Fig 1B). The IL-34 mRNA levels, as assessed by real-time RT-PCR, apparently increased in MRPTEpiC over time up to 24 h after CP stimulation (6 h: $P = 0.078$ vs. 0 h; 12 h: $P < 0.05$ vs. 0 h; 24 h: $P < 0.01$ vs. 0 h, $P < 0.05$ vs. 6 h, and $P < 0.05$ vs. 12 h; Fig 1C), and the IL-34 protein levels in the culture supernatants, as assessed by ELISA, were also increased after stimulation in a time-dependent manner up to 24 h (6 h: 7.2 ± 5.2 pg/mL, $P = 0.2883$ vs. 0 h; 12 h: 36.0 ± 5.6 pg/mL, $P < 0.05$ vs. 0 h, $P < 0.05$ vs. 6 h; 24 h: 69.5 ± 9.8 pg/mL, $P < 0.01$ vs. 0 h, $P < 0.01$ vs. 6 h, $P < 0.05$ vs. 12 h; Fig 1E). Similar expression patterns were detected in both the CSF-1 mRNA levels in MRPTEpiC (Fig 1D), and the CSF-1 protein levels in the culture supernatants (Fig 1F).

As shown in S1A Fig, the expression of cFMS in MRPTEpiC was detected by WB analysis at 12 h after stimulation with CP, and the densitometric analysis of cFMS in MRPTEpiC

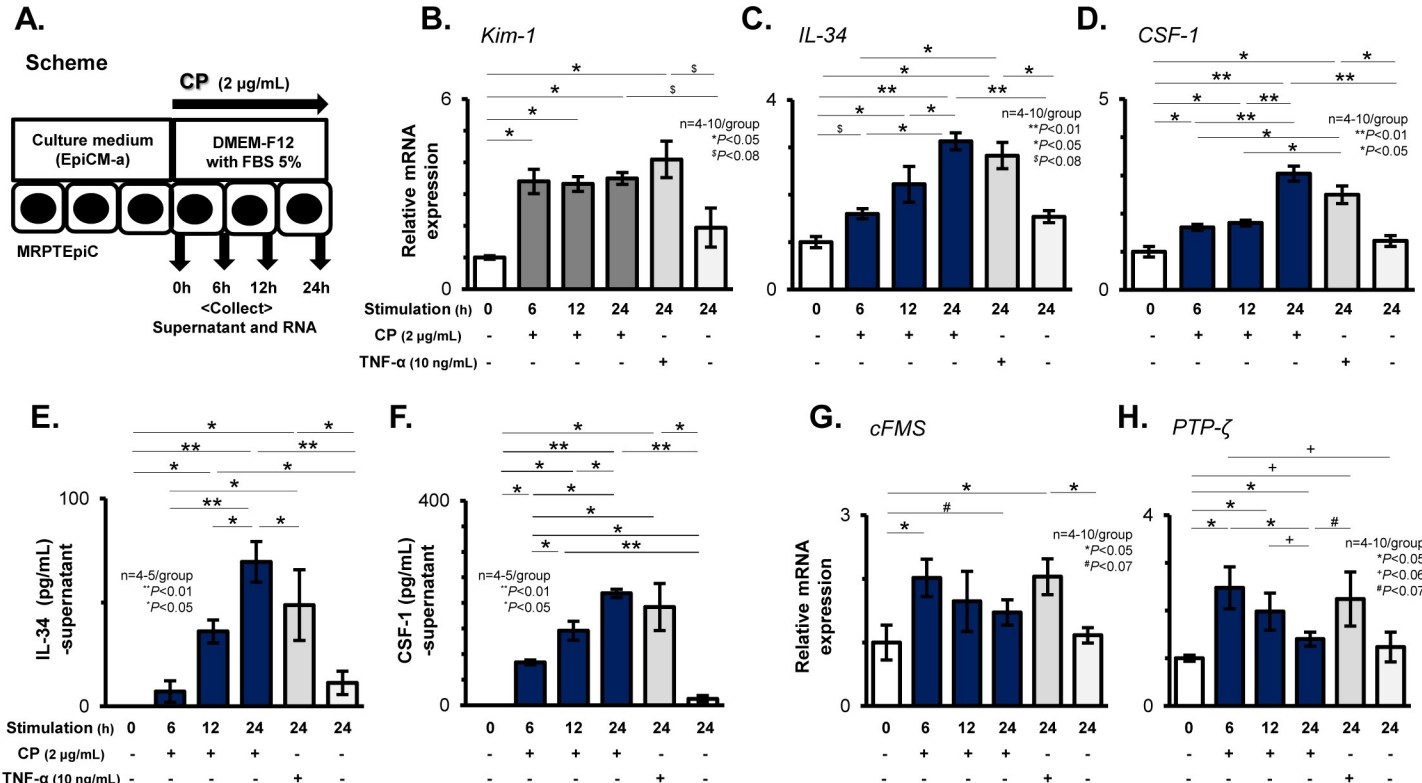

**Fig 1. Expression of IL-34 and its two receptors on CP-stimulated MRPTEpiC.** Scheme for the *in vitro* analysis using cultured MRPTEpiC stimulated with CP (**A**). MRPTEpiC and the culture supernatants were harvested after stimulation with CP (2 μg/mL) for 0 to 24 h. Real-time RT-PCR for Kim-1 (**B**), IL-34 (**C**), and CSF-1 (**D**) in MRPTEpiC after stimulation with CP or TNF-α for 0 to 24 h. IL-34 and CSF-1 protein levels evaluated by ELISA in the culture supernatants of MRPTEpiC after stimulation with CP or TNF-α for 0 to 24 h (**E** and **F**). Real-time RT-PCR for cFMS (**G**) and PTP-ζ (**H**) in MRPTEpiC after stimulation with CP or TNF-α for 0 to 24 h. In the analysis of real-time RT-PCR, the values were normalized to the β-actin transcript, and are expressed as a relative ratio. Data are expressed as the mean ± SEM. The Mann-Whitney U test was used for statistical analysis.

showed elevations in the values at 12 after CP stimulation when compared to the values of unstimulated cells (S1B Fig). Additionally, the transcripts of cFMS in MRPTEpiC were also up-regulated at the early phase after CP stimulation (6 h: P < 0.05 vs. 0 h), and a similar significant increase in cFMS transcripts was also seen after stimulation with TNF-α for 24 h (Fig 1G). Regarding PTP-ζ expression in CP-stimulated MRPTEpiC, although WB analysis showed only faint bands for PTP-ζ (S1A Fig), the densitometric analysis of PTP-ζ expression showed elevated values at 12 and 24 h after stimulation (S1C Fig), and the mRNA levels for PTP-ζ were significantly up-regulated from the early phase after stimulation with CP (6 h: P < 0.05 vs. 0 h; 12 h: P < 0.05 vs. 0h), and after 24 h of stimulation with TNF-α (Fig 1H).

## IL-34 generated from damaged TECs after CP stimulation promoted Mø proliferation

To confirm whether IL-34 generated from damaged TECs after CP stimulation directly induces Mø proliferation, we conducted *in vitro* experiments. As shown in Fig 2A, RAW 264, recognized as mouse Møs, were cultured for 12 h with a medium or the culture supernatant of MRPTEpiC stimulated with CP (2 μg/mL). The media for culturing RAW 264 contained vehicle, CP, rIL-34 (500 pg/mL), or anti-IL-34 Ab (1000 pg/mL). CP-stimulated MRPTEpiC were treated with vehicle or anti-IL-34 Ab (1000 pg/mL) for 24 h, followed by the collection of the supernatant for culturing RAW 264. Fig 2B shows the proliferation of the RAW 264 as

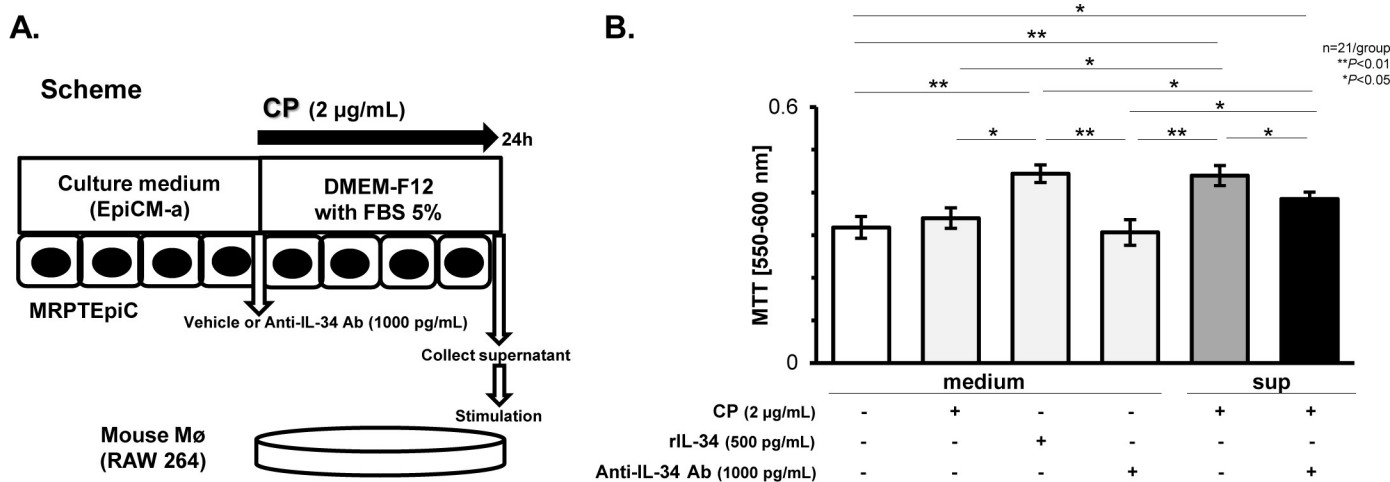

**Fig 2. Mø proliferation induced by the culture supernatants of MRPTEpiC stimulated with CP.** Scheme for the analysis. Mouse Møs (RAW 264) were cultured with medium or the culture supernatant of MRPTEpiC stimulated with CP (2 μg/mL) for 24 h. The medium for culturing RAW 264 contained vehicle, CP, rIL-34 (500 pg/mL), or anti-IL-34 Ab (1000 pg/mL). CP-stimulated MRPTEpiC were treated with vehicle or anti-IL-34 Ab (1000 pg/mL), followed by the collection of the supernatant to culture RAW 264 (**A**). MTT assay for Mø proliferation among the study groups (**B**). Data are expressed as the mean ± SEM. The Mann-Whitney U test was used for statistical analysis.

evaluated by the MTT assay among the study groups. RAW 264 stimulated with rIL-34-containing medium showed significantly increased proliferation when compared to the RAW 264 cultured in medium only (baseline; P < 0.01), in CP-containing medium (P < 0.05), or in anti-IL-34 Ab-containing medium (P < 0.01). In addition, the supernatant of CP-stimulated MRPTEpiC significantly increased the proliferation of the RAW 264 (P < 0.01 vs. baseline), as did the rIL-34-containing medium; this increase in proliferation was significantly suppressed by the supernatant of MRPTEpiC treated with CP and anti-IL-34 Ab (P < 0.05 vs. rIL-34-containing medium; P < 0.05 vs. the supernatant of CP-stimulated MRPTEpiC).

## CP-N induces intra-renal expression of IL-34 and its two receptors in mice

Representative photos of immunohistochemical staining for IL-34 among the study groups are shown in Fig 3A–3C. IL-34 was expressed in damaged TECs in CP-N mice (Fig 3B), and the positivity per HPF was significantly higher in the CP+V mice than in the NC mice (P < 0.05; Fig 3D). The protein levels of IL-34 in kidney tissues were also significantly higher in the CP +V mice than in the NC mice (P < 0.05; Fig 3E), which was compatible with the results for the renal cortical transcripts of IL-34 (S2A Fig). Meanwhile, the IL-34 expression on TECs was attenuated in the anti-IL-34 Ab-treated CP-N mice (Fig 3C), and the positivity was significantly lower in the CP+anti-IL-34 mice than in the CP+V mice (P < 0.05; Fig 3D). In addition, treatment with anti-IL-34 Ab significantly suppressed the up-regulation of IL-34 transcripts (P < 0.05 vs. CP+V; S2A Fig), and reduced the IL-34 protein levels (P = 0.061 vs. CP+V; Fig 3E) in the kidneys of CP-N mice.

Regarding the intra-renal IL-34 receptors in CP-N mice, the expression of cFMS was apparent in the CP+V group by WB analysis (Fig 3F), and the mRNA levels for cFMS were significantly higher in the CP+V mice (P < 0.05 vs. NC; S2B Fig). Similarly, PTP-ζ expression was apparent in the CP+V group in WB analysis (Fig 3F), and the densitometric values of PTP-ζ expression were significantly increased in the CP+V mice when compared to the NC mice (P < 0.01; Fig 3H). On the other hand, the CP+anti-IL-34 Ab group of mice showed faint expression of cFMS and PTP-ζ in the WB analysis (Fig 3F). The significant elevations in the

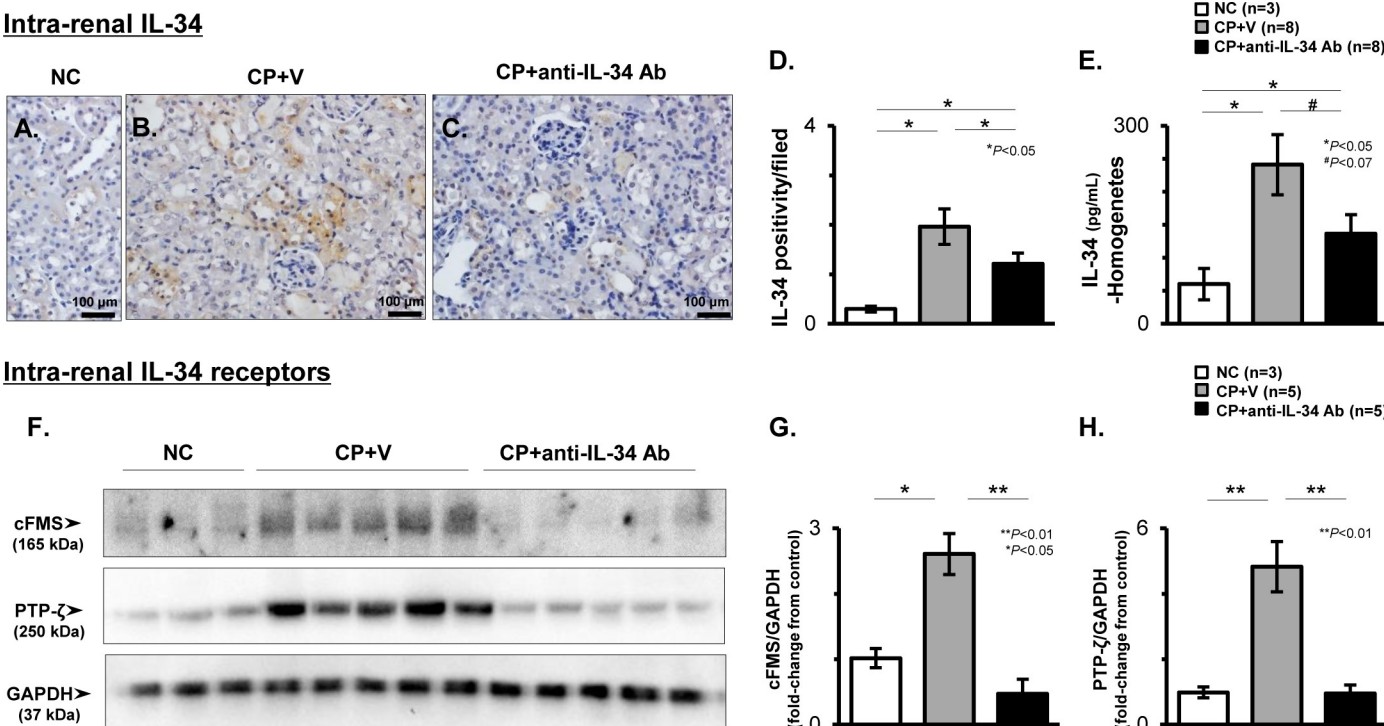

**Fig 3. Intra-renal expression of IL-34 and its two receptors in CP-N mice.** Representative photos of IL-34 expression on TECs identified by immunostaining among the NC (**A**), CP+V (**B**), and CP+anti-IL-34 Ab (**C**) groups of mice. Original magnification, 400×. Graph of the IL-34 positivity among the study groups (**D**). IL-34 protein levels in homogenate kidney tissues analyzed by ELISA among the study groups (**E**). Representative WB analysis for cFMS, PTP-ζ, and GAPDH (**F**). Densitometric analysis of WB for cFMS (**G**) and PTP-ζ (**H**). The values shown are the values after normalization to GAPDH expression, and they are depicted as the relative ratio of cFMS or PTP-ζ to GAPDH. Data are expressed as the mean ± SEM. The Mann-Whitney U test was used for statistical analysis.

densitometric values and the transcripts for cFMS in CP-N mice (P <0.05 vs. NC, respectively; Fig 3G and S2B Fig) were significantly suppressed in the anti-IL-34 Ab-treated mice (P <0.01 vs. CP+V; Fig 3G, P <0.05, vs. CP+V; S2B Fig). In addition, the up-regulation of the densitometric values of PTP-ζ (P < 0.01 vs. NC; Fig 3H) or elevated transcripts for PTP-ζ in the CP +V mice was significantly suppressed in the CP+anti-IL-34 Ab mice (P <0.01 vs. CP+V; Fig 3H, P < 0.05 vs. CP+V; S2C Fig).

## Effects of anti-IL-34 Ab on renal dysfunction and renal histological findings in CP-N mice

As shown in Fig 4A, the significant elevation of serum IL-34 levels in the CP-N mice was significantly suppressed in the anti-IL-34 Ab-treated mice (CP+anti-IL-34 Ab: 37.4 pg/mL; CP +V: 70.1 pg/mL; P < 0.05). No mortality was observed in any of the groups throughout the study period. The body weight (BW) at the time of sacrifice was significantly lower in the CP +V mice and CP+anti-IL-34 Ab mice than in the NC mice (P < 0.05, respectively; Fig 4B), but no significant difference was detected among the two groups. Meanwhile, the kidney weight (KW) in CP-N mice was significantly decreased among the anti-IL-34 Ab-treated mice (P < 0.01 vs. CP+V; S3 Fig). The KW-to-BW (KW/BW) ratio was also significantly lower in the CP+anti-IL-34 Ab mice than in the CP+ V mice (P < 0.01; Fig 4C). With regard to renal function, the s-Cr levels were significantly higher in the CP+V mice than in the NC mice (P < 0.01; Fig 4D), reflecting the marked elevation of Kim-1 mRNA levels in the CP-N mice (P < 0.01 vs NC; Fig 4E). The CP+anti-IL-34 Ab mice showed a significant reduction in the

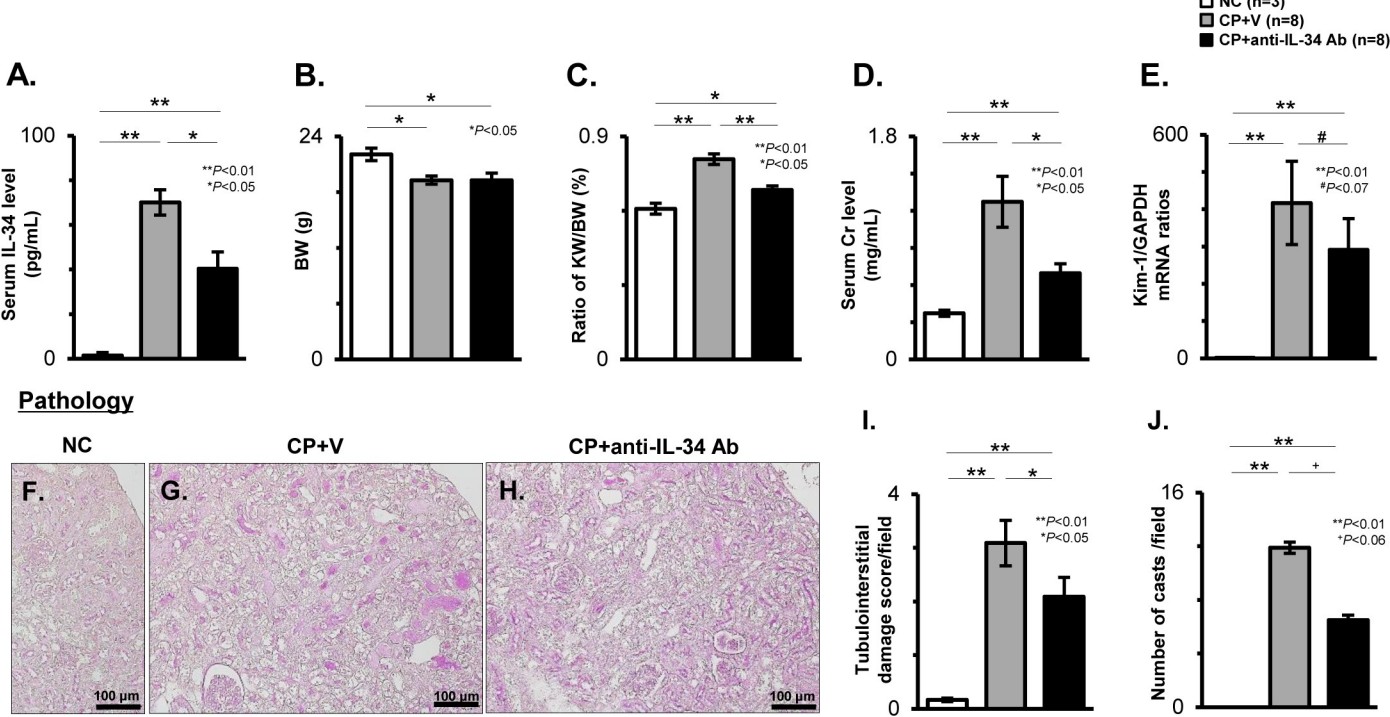

**Fig 4. Biochemical parameters and renal histological findings in CP-N mice.** The serum IL-34 levels as evaluated by ELISA (**A**), the body weight (BW) (**B**), and the ratios of the kidney weight (KW) to the BW (**C**) among the study groups. The serum Cr levels determined by the Cr assay kit (**D**) among the study groups. The mRNA levels for Kim-1 among the study groups (**E**). Values were normalized to the GAPDH transcript, and are expressed as the relative ratio. Representative photos of kidney tissues stained with PAS in a NC mouse (**F**), CP+V mouse (**G**), and CP+anti-IL-34 Ab mouse (**H**). Original magnification, 200×. Quantification of the tubulointerstitial damage score per field (**I**), and the number of casts per field (**J**) among the study groups. Data are expressed as the mean ± SEM. The Mann-Whitney U test was used for statistical analysis.

levels of s-Cr (CP+anti-IL-34 Ab: 0.7 mg/mL; CP+V: 1.3 mg/mL; P < 0.05), and suppression of Kim-1 transcripts (P = 0.069) when compared to the CP+V mice (Fig 4D and 4E).

Regarding renal pathology, representative photos of PAS staining from the study groups are shown in Fig 4F–4H. Compared to the NC mice (Fig 4F), the tubulointerstitial damage score and number of luminal casts/field in the CP+V mice (Fig 4G) were significantly higher (Fig 4I and 4J; P < 0.01, respectively). Treatment with anti-IL-34 Ab significantly ameliorated the damage score (Fig 4I; P < 0.05), and appeared to reduce the number of casts (Fig 4J; P = 0.056) in CP-N mice (Fig 4H).

### Effects of anti-IL-34 Ab on intra-renal Mø infiltration and TEC apoptosis in CP-N mice

Mø accumulation in the renal cortex of animals from each group was evaluated by IHC of F4/80. The CP+V mice showed a significant increase in the number of F4/80-positive Møs when compared to the NC mice (P < 0.05; Fig 5A, 5B, and 5D). Treatment with anti-IL-34 Ab significantly attenuated the number of F4/80-positive Møs in CP-N (P < 0.05 vs. CP+V; Fig 5C and 5D). To evaluate TEC apoptosis in the renal cortex, IHC staining of TUNEL or caspase-3 was performed. In the CP+V mice, the number of TUNEL-positive TECs was significantly increased when compared to that in the NC mice (P < 0.01; Fig 5E, 5F, and 5H). This increased number of apoptotic cells was significantly attenuated by the administration of anti-IL-34 Ab (P < 0.05; Fig 5G and 5H). Similarly, the increased number of caspase-3-positive cells in CP-N was significantly attenuated by anti-IL-34 Ab treatment (P < 0.05; Fig 5I–5L).

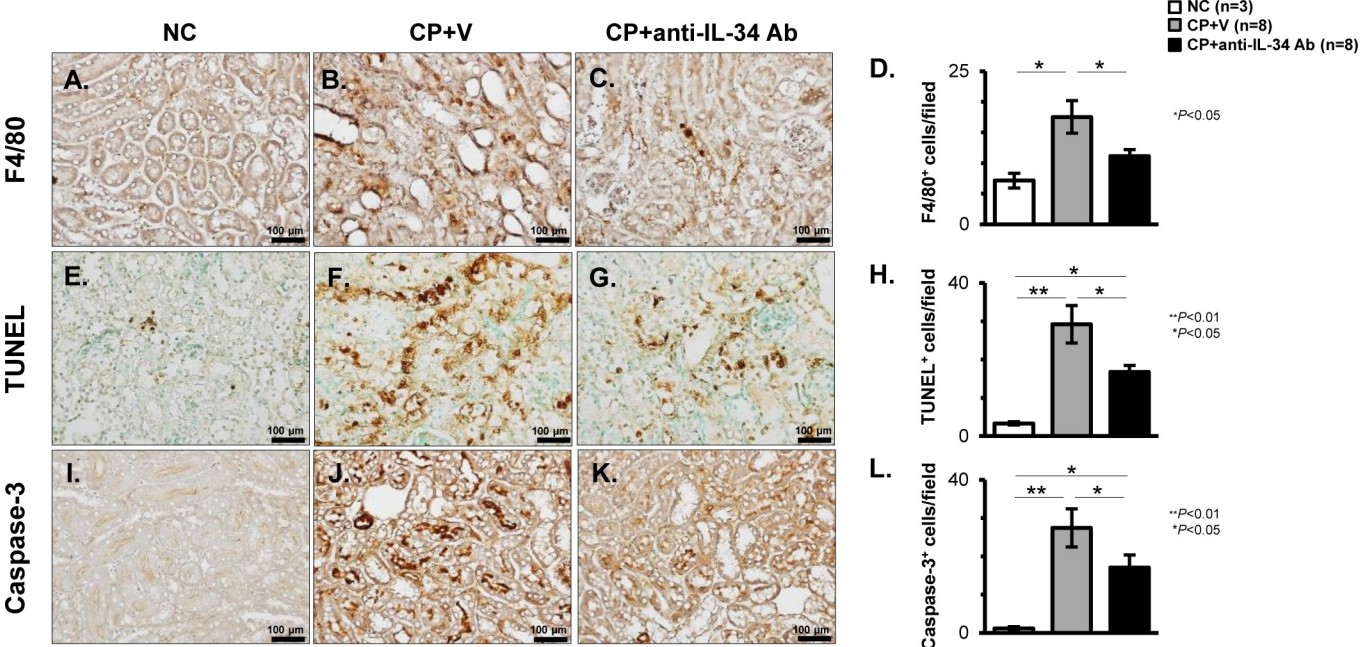

**Fig 5. Intra-renal Mø proliferation and TEC apoptosis in CP-N mice.** Representative photos of F4/80-positive Mø infiltration in the tubular interstitium, TUNEL-positive apoptotic cells in TECs, and caspase-3-positive cells in TECs identified by immunostaining among the NC (**A**, **E**, and **I**), CP+V (**B**, **F** and **J**), and CP+anti-IL-34 Ab (**C**, **G**, and **K**) groups of mice. Original magnification, 200×. Quantitative evaluation of the IHC for F4/80 (**D**), TUNEL (**H**), and caspase-3 (**L**) among the study groups. Data are expressed as the mean ± SEM. The Mann-Whitney U test was used for statistical analysis.

## Effects of anti-IL-34 Ab on the expression of genes encoding chemokines, apoptosis-regulatory molecules, and proinflammatory cytokines in CP-N mice

As shown in Fig 6A and 6B, the mRNA levels of chemokines, including MCP-1 (also known as CCL2) and MIP-1α (also known as CCL3), were significantly higher in the CP+V mice than in the NC mice (MCP-1: P < 0.05; MIP-1α: P < 0.01). Anti-IL-34 Ab treatment significantly suppressed the up-regulation of MIP-1α transcripts in CP-N (P < 0.05; Fig 6B), and the anti-IL-34 Ab-treated CP-N mice tended to have lower transcript levels for MCP-1 when compared to the vehicle-treated CP-N mice (Fig 6A). Regarding apoptosis-regulatory molecules, the levels of pro-apoptotic Bax transcripts were significantly higher in the CP+V mice than in the NC mice (P < 0.01), and this increase was significantly attenuated by anti-IL-34 Ab treatment (P < 0.05; Fig 6C). The expression level of Bcl-2, which encodes an anti-apoptotic protein, was significantly lower in the CP+V mice than in the NC mice and the CP+anti-IL-34 Ab mice (P < 0.05, respectively; Fig 6D). Regarding proinflammatory cytokines, the levels of transcripts for TNF-α, IL-6, or IL-1β were higher in the CP+V mice than in the NC mice (Fig 6E–6G). The increased expression of these genes was apparently lower in the anti-IL-34 Ab-treated CP-N mice than in the vehicle-treated CP-N mice (TNF-α: P = 0.052; IL-6: P < 0.05; IL-1β: P = 0.056; Fig 6E–6G). In addition, the elevation in the levels of the transcripts for IL-10 was comparable between the CP+V mice and the CP+anti-IL-34 Ab mice (Fig 6H).

## Effects of anti-IL-34 Ab on intra-renal M1-like and M2-like Mø accumulation in CP-N mice

Representative FACS plots with gating for the F4/80$^+$TNFα$^+$ or IL-10$^+$ population in CP+V mice and CP+anti-IL-34 Ab mice are shown in Fig 7A and 7B. We stained single-cell

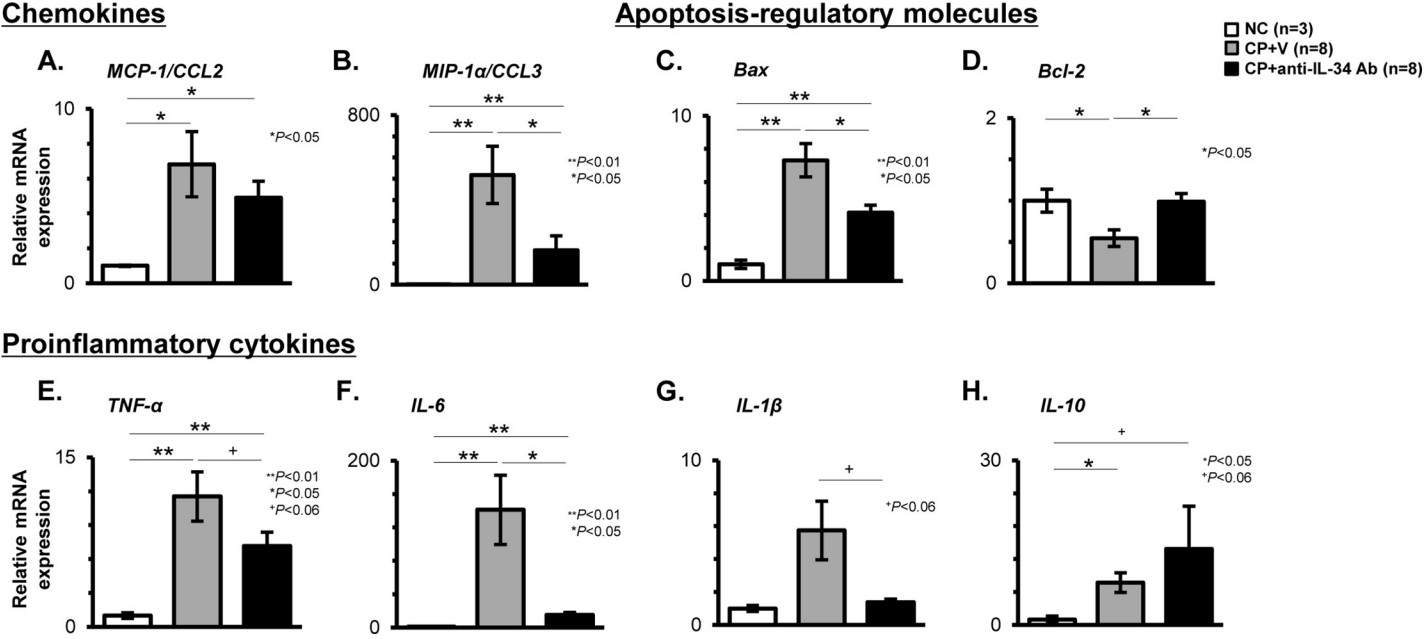

**Fig 6. Intra-renal expression of chemokines, apoptosis-regulatory molecules, and proinflammatory cytokines in CP-N mice.** Real-time RT-PCR for genes encoding MCP-1/CCL2 (**A**), MIP-1α/CCL3 (**B**), Bax (**C**), Bcl-2 (**D**), TNF-α (**E**), IL-6 (**F**), IL-1β (**G**), and IL-10 (**H**) among the NC, CP+V, and CP+anti-IL-34 Ab groups of mice. Values were normalized to the GAPDH transcript, and are expressed as the relative ratio. Data are expressed as the mean ± SEM. The Mann-Whitney U test was used for statistical analysis.

suspensions from both kidneys of each mouse. The CD11b⁺F4/80⁺ cell population was gated first, then the F4/80⁺TNFα⁺ or IL-10⁺ cell population was gated. We broadly recognized the F4/80⁺TNFα⁺ cells as cyto-destructive Møs (M1-like), and the F4/80⁺IL-10⁺ cells as cyto-protective Møs (M2-like). As shown in Fig 7C, the increased number of intra-renal cyto-destructive Møs in the CP+V mice was suppressed in the CP+anti-IL-34 Ab mice (P = 0.072). Meanwhile, the accumulation of cyto-protective Møs in kidney was comparable between the two groups (Fig 7D).

## Effects of anti-IL-34 Ab on cellular injury and apoptosis in cultured TECs after stimulation with CP

Cultured MRPTEpiC were stimulated with or without CP, followed by treatment with vehicle or anti-IL-34 Ab for 12 or 24 h (Fig 8A). As shown in Fig 8B, the time-dependent up-regulation of IL-34 transcripts in CP-stimulated MRPTEpiC was significantly suppressed by anti-IL-34 Ab treatment for 24 h (P < 0.05). Additionally, treatment with anti-IL-34 Ab significantly suppressed the up-regulation of Kim-1 mRNA levels in MRPTEpiC after stimulation with CP for 12 h (P < 0.05) and 24 h (P < 0.05; Fig 8C). The increased cytotoxicity in CP-stimulated MRPTEpiC, as evaluated by the LDH assay (S4A–S4D Fig), was also apparently suppressed by anti-IL-34 Ab treatment for 12 h (P < 0.05; S4C Fig) and 24 h (P = 0.069; S4D Fig). Moreover, the significant up-regulation of Bax transcripts was significantly suppressed by anti-IL-34 Ab treatment in MRPTEpiC stimulated with CP for 24 h (P < 0.05; Fig 8D), whereas the Bcl-2 transcripts in CP-stimulated MRPTEpiC were comparable among the study groups (Fig 8E). Analysis by WB showed the apparent expression of Bax in MRPTEpiC stimulated with CP for 24 h (Fig 8F), and the densitometric value of Bax in CP-stimulated MRPTEpiC was significantly decreased in the anti-IL-34 Ab-treated group after 24 h (P < 0.05; Fig 8G).

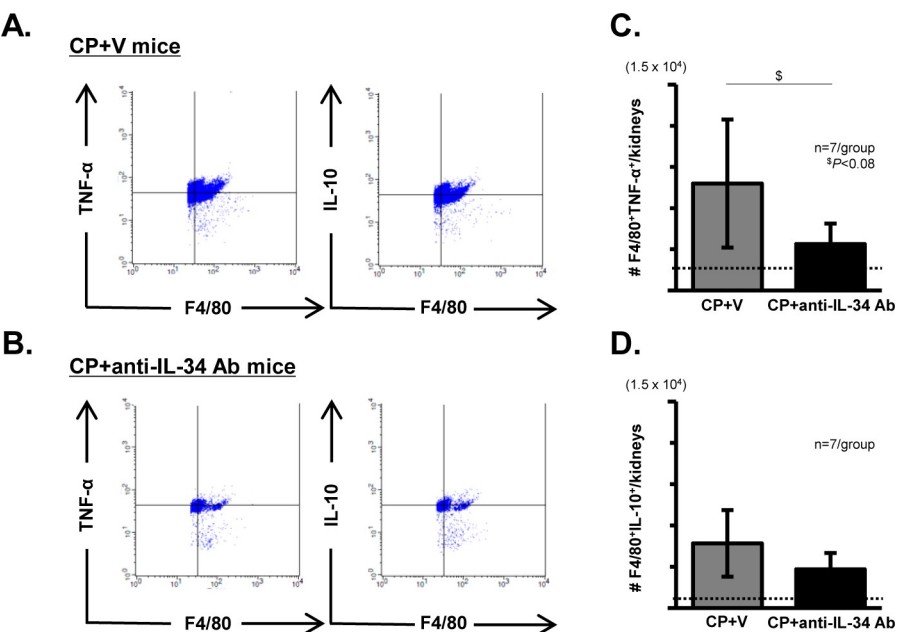

**Fig 7. Intra-renal cyto-destructive (M1-like) and cyto-protective (M2-like) Mø accumulation in CP-N mice.**
Single-cell suspensions from both kidneys of each mouse were stained. The CD11b$^+$F4/80$^+$ cell population was gated first, then the F4/80$^+$TNF$\alpha^+$ or IL-10$^+$ cell population was gated. Representative FACS plots with gating for F4/80$^+$TNF$\alpha^+$ or IL-10$^+$ cells in the CP+V (**A**) and CP+anti-IL-34 Ab mouse (**B**) groups. Graphs of the numbers of cyto-destructive Møs (F4/80$^+$TNF$\alpha^+$) (**C**) and cyto-protective Møs (F4/80$^+$IL-10$^+$) (**D**) in the kidneys of the mouse groups. Data are expressed as the mean ± SEM. Broken lines indicate the data for NC mice (n = 3). The Mann-Whitney U test was used for statistical analysis.

## Effects of anti-IL-34 Ab on the expression of CSF-1 in CP-N mice and CP-induced damaged TECs

To determine whether CSF-1 is mediated by anti-IL-34 Ab treatment, we evaluated the CSF-1 expression between the study groups. As shown in representative photos of IHC for CSF-1 in CP-N mice (Fig 9A–9C), CSF-1 was clearly expressed on TECs in the CP+V mice and the CP+anti-IL-34 Ab mice, but the positivity was comparable among the two groups (Fig 9D). Also, the serum CSF-1 levels were not different between the two groups (Fig 9E). In damaged MRPTEpiC after stimulation with CP for 24 h, the up-regulation of CSF-1 transcripts did not differ between the groups with or without anti-IL-34 Ab treatment (Fig 9F), and the elevation of protein levels in the culture supernatant was also comparable between the groups with or without anti-IL-34 Ab treatment (Fig 9G).

## Effects of anti-IL-34 Ab on cell signaling pathways in CP-N mice and CP-induced damaged TECs

To evaluate the involvement of IL-34 on the mitogen-activated protein kinase (MAPK) pathway and the phosphoinositide 3-kinase (PI3K)-Akt signaling pathway, which regulate cellular processes, such as cell proliferation, apoptosis, and differentiation [30, 38, 43], the phosphorylation of ERK1/2, a key kinase in MAPK signaling, and Akt in CP-N mice and damaged MRPTEpiC after stimulation with CP for 6 h were analyzed by WB. As shown in Fig 10A, the renal cortical expression of p-ERK1/2 was apparent in the CP+V mice, and the increased expression level was significantly attenuated by anti-IL-34 Ab treatment in CP-N mice (P < 0.05; Fig 10B). *In vitro*, the expression pattern of p-ERK1/2 was similar to that in CP-N

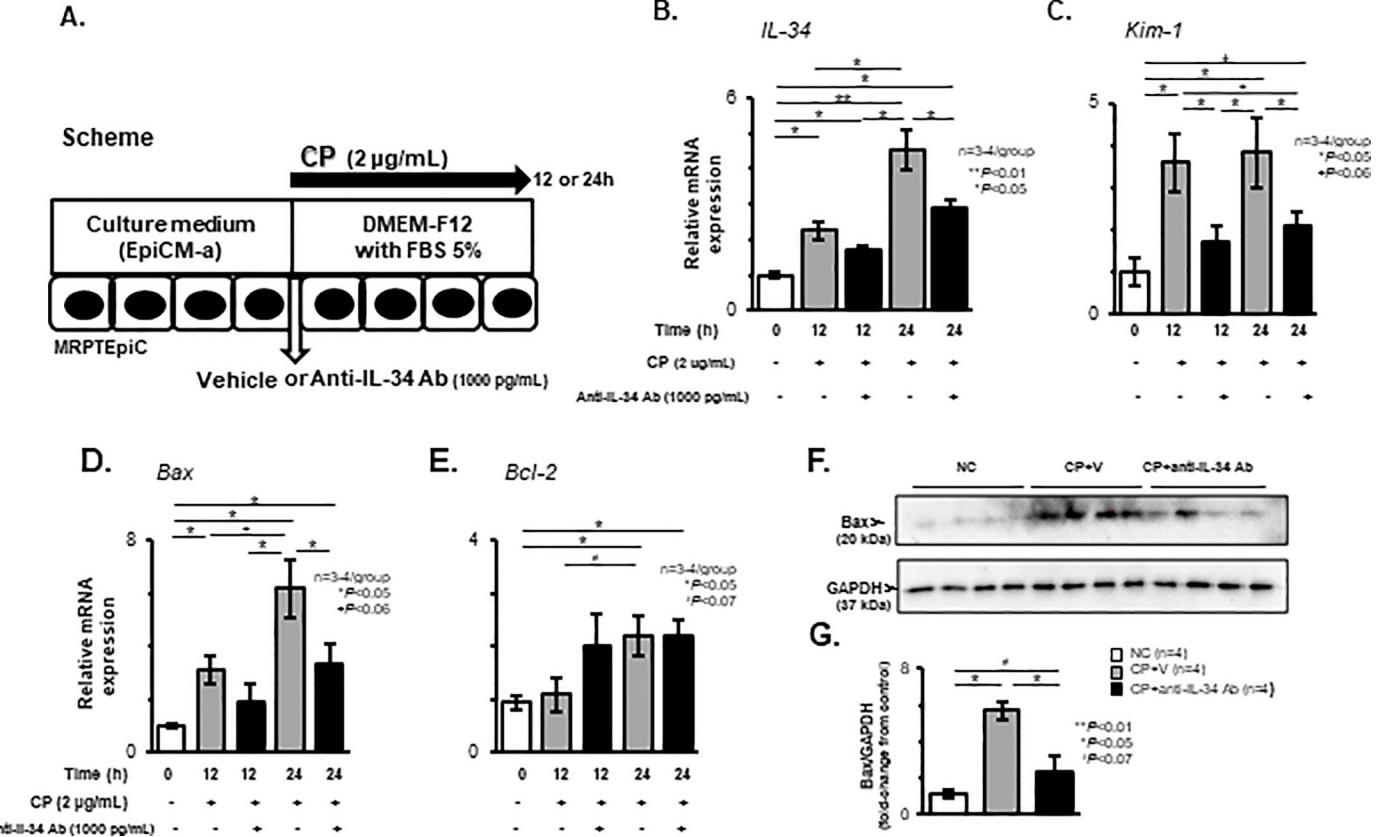

**Fig 8. Cellular injury and apoptosis in CP-stimulated MRPTEpiC.** Scheme for the *in vitro* analysis. Cultured MRPTEpiC were stimulated with or without CP (2 μg/mL), followed by treatment with vehicle or anti-IL-34 Ab (1000 pg/mL) for 12 or 24 h (**A**). The IL-34 (**B**) and Kim-1 (**C**) mRNA expression levels in TECs, as assessed by real-time RT-PCR, among the study groups. Real-time RT-PCR for Bax (**D**) and Bcl-2 (**E**) in CP-stimulated MRPTEpiC with or without anti-IL-34 Ab treatment for 0 to 24 h. Values were normalized to the GAPDH transcript, and are expressed as the relative ratio. Representative WB analysis for Bax and GAPDH (**F**). Densitometric analysis of WB for Bax. The values shown are the values after normalization to GAPDH expression, and they are depicted as the relative ratio of Bax to GAPDH (**G**). Data are expressed as the mean ± SEM. The Mann-Whitney U test was used for statistical analysis.

mice (Fig 10C), and densitometric analysis showed that an apparent elevation of p-ERK1/2 in the vehicle-treatment group (P < 0.05 vs. NC) was significantly suppressed by treatment with anti-IL-34 Ab in CP-stimulated MRPTEpiC (P < 0.05; Fig 10D). Regarding the phosphorylation of Akt, both CP-N mice and CP-stimulated damaged MRPTEpiC showed faint bands for p-Akt, and no densitometric difference among the study groups was detected in the *in vivo* and *in vitro* analyses (S5 Fig).

## Discussion

To our knowledge, the present study proved for the first time that the blocking of IL-34 with nAb attenuated the progression of CP-N, and we uncovered the following points: (1) the expression of IL-34 and its two receptors, cFMS and PTP-ζ, was increased in the kidneys of CP-N mice and in cultured TECs after stimulation with CP; (2) *in vivo*, the administration of anti-IL-34 Ab reduced the intra-renal expression of IL-34 and its two receptors in CP-N mice, which resulted in the attenuation of CP-induced AKI without any adverse effects; (3) anti-IL-34 Ab-treated CP-N mice showed a reduction of intra-renal Mø infiltration, especially of cyto-destructive M1-like Møs, with decreased expression of chemokines and proinflammatory cytokines; (4) treatment with anti-IL-34 Ab suppressed TEC apoptosis in CP-N mice with the

## Intra-renal CSF-1 expression

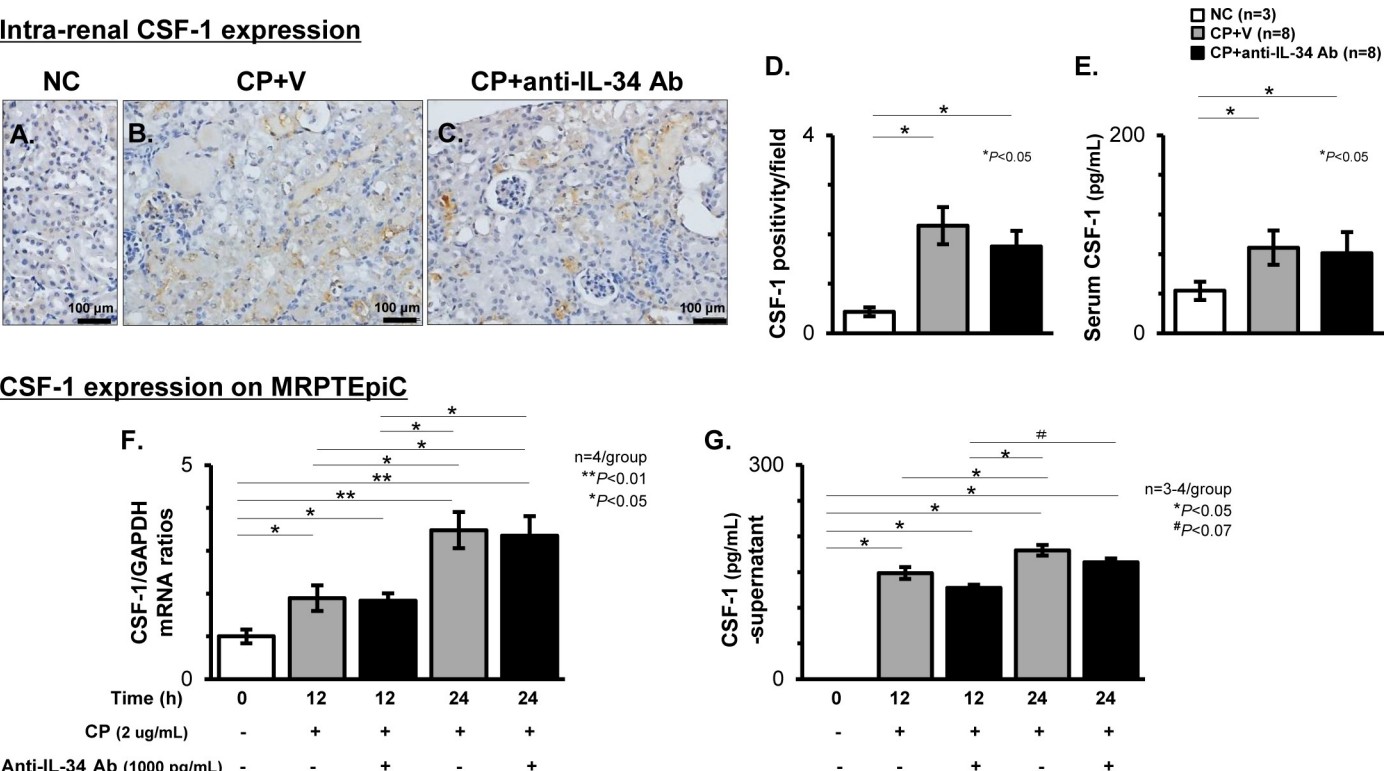

## CSF-1 expression on MRPTEpiC

**Fig 9. Expression of CSF-1 in CP-N mice and CP-stimulated MRPTEpiC.** Representative photos of the intra-renal CSF-1 expression on TECs identified by immunostaining among the NC (**A**), CP+V (**B**), and CP+anti-IL-34 Ab (**C**) groups of mice. Original magnification, 400×. Graph of CSF-1 positivity per field among the study groups (**D**). The serum CSF-1 levels evaluated by ELISA among the study groups (**E**). Real-time RT-PCR for CSF-1 in CP-stimulated MRPTEpiC with or without anti-IL-34 Ab treatment for 0 to 24 h (**F**). Values were normalized to the GAPDH transcript, and are expressed as the relative ratio. The CSF-1 protein levels evaluated by ELISA in the culture supernatants of MRPTEpiC after CP stimulation with or without anti-IL-34 Ab treatment for 0 to 24 h (**G**). Data are expressed as the mean ± SEM. The Mann-Whitney U test was used for statistical analysis.

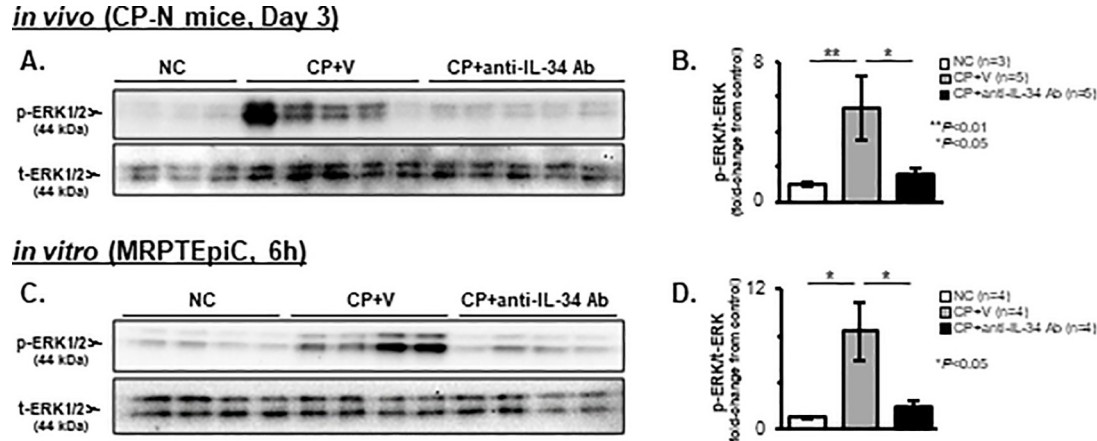

**Fig 10. CP-induced ERK1/2 phosphorylation in CP-N mice and CP-stimulated MRPTEpiC.** Representative WB analysis for p-ERK1/2 and t-ERK1/2 in renal cortical tissues after stimulation with CP for 72 h among the NC, CP+V, and CP+anti-IL-34 Ab groups of mice (**A**). Densitometric analysis of WB for p-ERK1/2 among the study groups (**B**). Representative WB analysis for p-ERK1/2 and t-ERK1/2 in MRPTEpiC after stimulation with CP for 6 h among the NC, CP+V, and CP+anti-IL-34 Ab groups (**C**). Densitometric analysis of WB for p-ERK1/2 and t-ERK1/2 in MRPTEpiC among the study groups (**D**). The values shown are the values after normalization to the t-ERK1/2 expression, and they are depicted as the relative ratio of p-ERK1/2 to t-ERK1/2. Data are expressed as the mean ± SEM. The Mann-Whitney U test was used for statistical analysis.

down-regulation of apoptosis-regulatory molecules; (5) *in vitro*, the administration of anti-IL-34 Ab reduced the expression of IL-34 and its two receptors in CP-stimulated TECs, which inhibited TEC apoptosis via the suppression of apoptosis-regulatory molecules; (6) CSF-1 was not up-regulated in compensation for the decline of IL-34 in the *in vivo* and *in vitro* analyses of CP-N; and (7) treatment with anti-IL-34 Ab inhibited the phosphorylation of ERK1/2 in *in vivo* and *in vitro* models of CP-N.

In this study, CP-N up-regulated the expression of IL-34 in damaged TECs, as has been demonstrated in previous reports using mouse I/R or LN models [13, 15]. *In vitro*, IL-34 secreted from CP-stimulated MRPTEpiC fostered Mø proliferation, which was suppressed by the administration of anti-IL-34 Ab. IL-34 from TECs has a physiological property as a Mø mediator, and the anti-mouse IL-34 Ab, i.e., polyclonal sheep IgG produced in response to a mouse myeloma cell line, functioned as a nAb. Similar results were reported from an experiment using the same Ab in cultured TECs under hypoxia [13]. Furthermore, treatment with anti-IL-34 Ab inhibited the activation of IL-34 in CP-N mice, indicating that this nAb might be effective in the *in vivo* model. In this study, the anti-IL-34 Ab dosage was determined with reference to the manufacturer's data sheet, which stated that the neutralization dose (ND50) is typically 0.3 to 1.5 μg/mL in the presence of 100 ng/mL of IL-34. However, there are still unclarified points regarding the dynamics of IL-34 itself in contrast to CSF-1 [44]. Therefore, further examinations are necessary to identify the half-life of IL-34 as well as the proper dosage and interval of anti-IL-34 Ab treatment for the optimal therapeutic effect.

According to recent reviews [1, 45], the up-regulation of IL-34 is recognized to reflect the activity and/or severity of kidney disease. Indeed, we detected increased intra-renal IL-34 expression in CP-N mice and up-regulated IL-34 expression in CP-stimulated MRPTEpiC along with an elevation of Kim-1 expression. Furthermore, as was demonstrated previously [11, 13, 15], the present study showed that not only IL-34, but also its two receptors, cFMS and PTP-ζ, were up-regulated in the kidneys of CP-N mice, and the up-regulation of cFMS and PTP-ζ was also detected in cultured TECs after CP stimulation. Although the expression of receptors on TECs was not so remarkable, it is possible that IL-34 has an autocrine effect on TECs through the activation of cFMS or PTP-ζ when considering the high affinity of IL-34 to its receptors. Moreover, the nAb used in this study attenuated AKI and TEC damage with the down-regulation of tubular IL-34 expression. As such, TEC-derived IL-34 appears to promote CP-N by exerting its autocrine effect on TECs. Inhibition of the IL-34/cFMS or PTP-ζ axis may be effective for preventing the progression of CP-N. However, in contrast to our speculation, direct stimulation with only recombinant IL-34 did not result in TEC injury, as evaluated by the LDH assay; this led us to presume that the tubular autocrine effect of IL-34 may be restricted and harmless under physiological conditions unless some type of injury, such as CP-N, stimulates the IL-34-induced aggravating effect through the activation of its two receptors.

In general, Mø-based inflammatory responses are involved in the pathogenesis of AKI, including CP-N [20, 26, 27]. Intra-renal Mø infiltration is reported to be an exacerbation factors of CP-induced AKI [24, 26]. The present study also detected Mø accumulation in the injured kidneys of CP-N mice. Therefore, it is plausible that TEC injury in CP-N may be influenced by IL-34-mediated Møs. In a previous report using the mouse I/R model, IL-34 enhanced Mø-mediated TEC destruction and exacerbated AKI by promoting intra-renal Mø proliferation and enhancing the recruitment of BM-derived circulating monocytes into the inflamed kidneys through the activation of intra-renal chemokines [13]. In another study using LN mice with a genetic deletion of IL-34, the up-regulation of renal inflammatory cytokines that activate intra-renal chemokines was suppressed, leading to the attenuation of TEC injury via a reduction in intra-renal Mø accumulation [15]. Furthermore, in skin Langerhans

cells, IL-34 promoted inflammatory effects by inducing the expression of chemokines and pro-inflammatory mediators [3, 46]. Unfortunately, in the present study, we did not evaluate systemic factors, such as BM-generated circulating monocytes, or analyze immunocompetent cells other than Møs, e.g., neutrophils and lymphocytes. However, in the present study, anti-IL-34 Ab-treated CP-N mice, which exhibited improvements of their renal disorder, showed less infiltration of intra-renal Møs, and decreased expression of inflammatory cytokines and chemokines. Taken together, IL-34 appears to plays a key role in the development of Mø-mediated TEC damage in CP-N, and CP-N may be attenuated by suppressing the IL-34-induced intra-renal accumulation of Mø.

To our knowledge, the effect of IL-34 on Mø polarization in kidney diseases remains unclear. Unlike CSF-1, no skewing has been observed between M1 and M2 Møs in experimental kidney disease models [11, 13, 15, 39]. Other analyses using non-renal cell lines have demonstrated that IL-34-differentiated Møs may be polarized to the M2 phenotype, and exert anti-inflammatory effects [47, 48]. Additionally, Bezie et al. reported that T regulatory cells expressed IL-34, which modulated the Mø response so that IL-34-primed Møs potentiated the immune suppressive capacity of T regulatory cells [49]. These reports are inconsistent with our results, in which the blocking of IL-34 with nAb induced an anti-inflammatory response with the suppression of intra-renal TNF-α-positive Møs in CP-N mice. Usually, TNF-α is recognized to worsen inflammation and TEC injury in kidney diseases [1, 23, 50]. As such, it might be acceptable to consider the increased F4/80$^+$TNFα$^+$ cells detected in this study as cyto-destructive M1-like Møs rather than cyto-protective M2-like Møs, and therefore, we suggest that the suppression of M1-like Mø proliferation by blocking IL-34 might help attenuate TEC injury in CP-N mice. Nonetheless, both the previous analyses and our analyses have limitations. The definition of M1 or M2 Møs depended on the evaluation of surface markers only, and verification experiments based on their actual functions, i.e., whether IL-34-differentiated Møs exacerbate or alleviate TEC injury, were not performed. Moreover, the Møs accumulated in inflamed kidneys should be strictly classified as renal-resident Møs or BM-derived Møs, and the influence of IL-34 on each classification of Møs needs to be further clarified.

The inhibition of TEC apoptosis is essential in the therapeutic strategies against CP-N [23]. Bax expression on tubules is a critical pro-apoptotic factor for CP-N [23, 27]. In the present study, treatment with anti-IL-34 Ab suppressed TEC apoptosis with the degradation of Bax expression in CP-N mice, which might be attributable to the inhibition of IL-34-related mechanisms. Previously, the depletion of Møs due to the administration of clodronate ameliorated TEC apoptosis in experimental AKI models [51, 52]. Also, genetically deleting IL-34 in LN mice suppressed intra-renal Mø accumulation, which led to the prevention of TEC apoptosis [15]. Moreover, in an *in vitro* analysis of TECs co-cultured with TNF-α stimulated Møs, anti-IL-34 Ab treatment prevented TNF-α-induced TEC apoptosis via the suppression of Mø proliferation [15]. Thus, in this study, blocking IL-34 with nAb might have indirectly attenuated TEC apoptosis in CP-N mice via the inhibition of Mø-mediated cytotoxicity. Regarding the direct influence of IL-34 on TEC apoptosis, we focused on the cell signaling pathways that regulate apoptosis. The phosphorylation of ERK1/2 is reported to promote apoptosis with the up-regulation of Bax [53], whereas the PI3K-Akt pathway is reported to be related to anti-apoptotic functions [23]. Additionally, the binding of IL-34 to cFMS is reported to induce the phosphorylation of ERK or Akt [2, 54], and such phosphorylation has been confirmed through the activation of PTP-ζ [6, 55]. Taken together, anti-IL-34 Ab treatment appears to directly prevent CP-induced TEC apoptosis by inhibiting ERK1/2 phosphorylation through IL-34-induced signaling. However, unexpectedly, Akt phosphorylation was not apparent in the present study, and it remains unclear whether the signal from the binding of IL-34/cFMS or

IL-34/PTP-ζ is dominant for IL-34-induced ERK1/2 phosphorylation. Further analyses on the influence of IL 34 on cell signaling pathways are needed to establish the evidence.

In this study, we also examined CSF-1, a Mø mediator with reno-protective properties [1, 11]. Consistent with previous reports [13, 15], we demonstrated that CSF-1 in the damaged TECs and circulation did not compensate for the decline of IL-34 in CP-N. Consequently, we consider that the attenuation of CP-N in mice or cultured TECs treated with anti-IL-34 Ab was mainly due to the inhibition of IL-34-related effects rather than CSF-1-induced reno-protective effects. IL-34 has been reported to be a safer therapeutic target than CSF-1 [1, 45]. Indeed, IL-34-knockout mice did not express overt phenotypic abnormalities other than a decline in Langerhans cells and microglia [4]. Furthermore, it was reported that nAb for IL-34 is unlikely to cross the blood-brain barrier [1, 15]. Hence, the targeting of IL-34 might be a good therapeutic strategy for kidney diseases.

In conclusion, our data indicated that IL-34 secreted from damaged TECs binds to its receptors and aggravates CP-N by activating the autocrine/paracrine deteriorative effects of IL-34 on TECs. Treatment with anti-IL-34 Ab directly prevented CP-induced TEC apoptosis by inhibiting the phosphorylation of ERK1/2, and the blocking of IL-34 with nAb might have indirectly attenuated CP-N via the suppression of cytotoxic Mø proliferation. Thus, IL-34 has potential as a therapeutic target for kidney diseases, and the inhibition of IL-34 with nAb might have a reno-protective effect.

## Supporting information

**S1 Checklist.**
(PDF)

**S1 Fig. The expression of two receptors for IL-34 in cultured TECs after stimulation with CP.** Representative WB analysis for cFMS, PTP-ζ, and α-Tub expression on MRPTEpiC after CP stimulation or the addition of medium (Med) for 0 to 24 h (**A**). Densitometric analysis of WB for cFMS (**B**) and PTP-ζ (**C**). The values shown are the values after normalization to α-Tub expression, and they are depicted as the relative ratio of cFMS or PTP-ζ to α-Tub.
(PDF)

**S2 Fig. The mRNA expression levels of intra-renal IL-34 and its two receptors in CP-N mice.** Real-time RT-PCR for IL-34 (**A**), C-FMS (**B**), and PTP-ζ (**C**) among the NC, CP+V, and CP+anti-IL-34 Ab groups of mice. Values were normalized to the GAPDH transcript, and are expressed as the relative ratio. Data are expressed as the mean ± SEM. The Mann-Whitney U test was used for statistical analysis.
(PDF)

**S3 Fig. Kidney weight in CP-N mice.** Kidney wight (KW) among the NC, CP+V, and CP+anti-IL-34 Ab groups of mice. Data are expressed as the mean ± SEM. The Mann-Whitney U test was used for statistical analysis.
(PDF)

**S4 Fig. Cell injury in CP-stimulated MRPTEpiC.** Cultured MRPTEpiC were stimulated with or without CP (2 μg/mL), followed by treatment with vehicle or rIL-34 (500 pg/mL) or anti-IL-34 Ab (1000 pg/mL) for 12 (**A**) or 24 h (**B**). Cytotoxicity in cultured TECs, evaluated by LDH assay, at the time of 12h (**C**) and 24h (**D**) were compared among the study groups. Values in the group without CP-stimulation and any treatment were recognized as control, and each value in the study groups was calculated using its control data. Data are expressed as the

mean ± SEM. The Mann-Whitney U test was used for statistical analysis.
(PDF)

**S5 Fig. CP-induced Akt phosphorylation in CP-N mice and CP-stimulated MRPTEpiC.**
Representative WB analysis for p-Akt and t-Akt in renal cortical tissues after stimulation with
CP for 72 h among the NC, CP+V, and CP+anti-IL-34 Ab groups of mice (**A**). Representative
WB analysis for p-Akt and t-Akt in MRPTEpiC after stimulation with CP for 6 h among the
NC, CP+V, and CP+anti-IL-34 Ab groups (**B**).
(PDF)

**S1 Table. Information of primers used for real-time RT-PCR (TaqMan) assay.**
(RTF)

**S1 Raw image. Images of all original blot and gel images.**
(PDF)

## Acknowledgments

We thank Ms. Tomoko Suzuki and Ms. Kumiko Ueda for their excellent technical assistance.

## Author Contributions

**Conceptualization:** Yukihiro Wada, Masayuki Iyoda, Kei Matsumoto.

**Data curation:** Yukihiro Wada, Kei Matsumoto, Taihei Suzuki, Shohei Tachibana, Nobuhiro
Kanazawa.

**Formal analysis:** Yukihiro Wada, Kei Matsumoto, Taihei Suzuki, Shohei Tachibana, Nobu-
hiro Kanazawa.

**Funding acquisition:** Yukihiro Wada.

**Investigation:** Yukihiro Wada, Masayuki Iyoda, Kei Matsumoto, Hirokazu Honda.

**Methodology:** Yukihiro Wada, Masayuki Iyoda, Kei Matsumoto, Hirokazu Honda.

**Writing – original draft:** Yukihiro Wada.

**Writing – review & editing:** Yukihiro Wada, Masayuki Iyoda, Hirokazu Honda.

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
