## [Decision Letter · Decision Letter 0]

10 Dec 2020

PONE-D-20-32498

Reno-protective effect of IL-34 inhibition on cisplatin-induced nephrotoxicity in mice

PLOS ONE

Dear Dr. Yukihiro Wada

Thank you for submitting your manuscript to PLOS ONE. The paper has been revised by a reviewer and by the Editor. After careful consideration, we feel that it has merit. The experiments are performed with scientific rigour, and the results are novel.

There are few minor requests that you are requested to accomplish. Therefore, we invite you to submit a revised version of the manuscript that addresses the points raised during the review process.

We look forward to receiving your revised manuscript.

Kind regards,

Benedetta Bussolati, MD, PhD

Academic Editor

PLOS ONE

Journal Requirements:

2)  To comply with PLOS ONE submissions requirements, please provide the method of euthanasia in the Methods section of your manuscript.

3) Please provide additional information about each of the cell lines used in this work, including any quality control testing procedures (authentication, characterisation, and mycoplasma testing). For more information, please see http://journals.plos.org/plosone/s/submission-guidelines#loc-cell-lines.

4) At this time, we request that you  please report additional details in your Methods section regarding animal care, as per our editorial guidelines:

(i) Please provide details of animal welfare (e.g., shelter, food, water, environmental enrichment)

(ii) Please describe the post-operative care received by the animals, including the frequency of monitoring and the criteria used to assess animal health and well-being.

Thank you for your attention to these requests.

5) To comply with PLOS ONE submission guidelines, in your Methods section, please provide additional information regarding your statistical analyses, including the threshold set for statistical significance. For more information on PLOS ONE's expectations for statistical reporting, please see https://journals.plos.org/plosone/s/submission-guidelines.#loc-statistical-reporting.

6) In the Methods section, please provide the specific sequences of the primers used in the PCR analysis conducted in your study.

7) PLOS ONE now requires that authors provide the original uncropped and unadjusted images underlying all blot or gel results reported in a submission’s figures or Supporting Information files. This policy and the journal’s other requirements for blot/gel reporting and figure preparation are described in detail at https://journals.plos.org/plosone/s/figures#loc-blot-and-gel-reporting-requirements and https://journals.plos.org/plosone/s/figures#loc-preparing-figures-from-image-files. When you submit your revised manuscript, please ensure that your figures adhere fully to these guidelines and provide the original underlying images for all blot or gel data reported in your submission. See the following link for instructions on providing the original image data: https://journals.plos.org/plosone/s/figures#loc-original-images-for-blots-and-gels.

8) Thank you for stating the following in your Competing Interests section: 

[No conflicts of interest are declared by the authors.].

Reviewers' comments:

Reviewer's Responses to Questions

**Comments to the Author**

1. Is the manuscript technically sound, and do the data support the conclusions?

Reviewer #1: Yes

2. Has the statistical analysis been performed appropriately and rigorously? 

Reviewer #1: Yes

3. Have the authors made all data underlying the findings in their manuscript fully available?

Reviewer #1: Yes

4. Is the manuscript presented in an intelligible fashion and written in standard English?

Reviewer #1: Yes

5. Review Comments to the Author

Reviewer #1: In the paper entitled “Reno-protective effect of IL-34 inhibition on cisplatin-induced nephrotoxicity in mice”, authors described the role and the modulation of IL-34 and its receptors during damage in an in vitro and in vivo model of renal damage, caused by cisplatin nephrotoxicity. Authors dissected the effect of the blocking of IL-34 using a blocking antibody on TEC and macrophages in vitro and in vivo.They demonstrated that in vivo, the use of anti-IL-34 ab reduced the expression of IL-34 itself and its receptors and displayed a beneficial effect on renal recovery after cisplatin damage. In particular, the treatment generated a reduction of tubulointerstitial injury, suppressed macrophage infiltration, and reduced apoptotic cell.

In vitro authors demonstrated that cisplatin treatment on TEC induced the expression of IL-34,that in turn promoted the progression of damage.

The conclusion is that IL-34 may be a potential therapeutic target, ant its inhibition may display renoprotective effects.

The paper is well designed and well explained and clear.

I would suggest some minor changes to improve the manuscript.

The “abstract” is too long, I suggest being more concise. Moreover, in the introduction section the part describing macrophages (M1 and M2) should be summarize.

In the discussion section, I would suggest to be more discursive and add sentence before the point list.

Minor point:

Page 14, line 20, please change “dedicated” with specific

6. PLOS authors have the option to publish the peer review history of their article (what does this mean?). If published, this will include your full peer review and any attached files.

Reviewer #1: No

---

## [Author Response · Author response to Decision Letter 0]

22 Dec 2020

December 22, 2020

Dr. Benedetta Bussolati

Academic Editor

PLOS ONE

RE: PONE-D-20-32498 

Reno-protective effect of IL-34 inhibition on cisplatin-induced nephrotoxicity in mice

Dear Dr. Bussolati:

 We appreciate the meticulous review of our study and are pleased to learn that our submission is of interest. We welcome the opportunity to submit a revised manuscript. All changes, including major revisions for the problems listed below and minor modifications for English grammar, are shown in red font in the revised manuscript. 

 We have addressed the concerns raised as follows:

Journal requirements

1. No changes are required regarding our financial disclosure.

2. If applicable, we recommend that you deposit your laboratory protocols in protocols.io to enhance the reproducibility of your results. Protocols.io assigns your protocol its own identifier (DOI) so that it can be cited independently in the future. 

 Because the experimental protocols are clearly detailed in the methods section, we did not prepare a separate specific laboratory protocol.

 We carefully read PLOS ONE's style requirements and prepared the revised manuscript and file names based on the instructions.

4. To comply with PLOS ONE submissions requirements, please provide the method of euthanasia in the Methods section of your manuscript.

 We added following information in the Method section. At day 3 (72 h after the CP injection), each mouse was anesthetized and sacrificed by exsanguination after the cardiac puncture; blood was collected by cardiac puncture and kidneys were collected. Renal tissue was divided; separate portions were snap-frozen in liquid nitrogen or fixed in 2% paraformaldehyde/phosphate-buffered saline for later use. All surgery was performed under anesthesia by pentobarbital (100 mg/kg), and all efforts were made to minimize suffering (Page 7, Line 1-7).

5. Please provide additional information about each of the cell lines used in this work, including any quality control testing procedures (authentication, characterisation, and mycoplasma testing).

 We carefully checked data sheet of cell lines used in the experiments, including MRPTEpiC and RAW 264. Then, we described additional information in the Method section (Page 11, Line 15-23). 

 According to the manufacturer’s (ScienCell Research Laboratories) data sheet, purchased MRPTEpiC (catalog number: M4100-57) were characterized by immunofluorescence with antibodies specific to cytokeratin-18, -19, and vimentin. The MRPTEpiC are negative for mycoplasma, bacteria, yeast, and fungi. MRPTEpiC are guaranteed to further culture under the condition provided by ScienCell Research Laboratories. 

 Also, resource number of RAW 264 provided by RIKEN Cell Bank is RBRC-RCB0535. Authentication of the RAW 264 was determined by isozyme analysis (Lot31: LD, NP), and chromosome diversion was following: Lot1: 39-160 (50): 39 (5), 40 (36), 41 (1), 78 (1), 80 (3). Furthermore, the RAW 264 are negative for mycoplasma, bacteria, yeast, and fungi.

6. At this time, we request that you please report additional details in your Methods section regarding animal care, as per our editorial guidelines:

(i) Please provide details of animal welfare (e.g., shelter, food, water, environmental enrichment)

 We described an additional information as follow: the mice were housed in the animal care facility of Showa University under standard conditions (25℃, 50% relative humidity, 12-hour dark/light cycle) with free access to food and water (Page 6, Line 10-12).

(ii) Please describe the post-operative care received by the animals, including the frequency of monitoring and the criteria used to assess animal health and well-being.

 Fundamentally, mice used in the present study were weighed daily and food-intake was monitored twice daily during the experiment (Page 6, Line 12-13). In addition, animal health and well-being were monitored at the point of 1 h after every procedure of IP during the experiment (Page 6, Line 23-Page 7, Line 1).

7. To comply with PLOS ONE submission guidelines, in your Methods section, please provide additional information regarding your statistical analyses, including the threshold set for statistical significance.

 In all the analyses, P values of < 0.05 were considered to be statistically significant. We described an additional explanation regarding statistical analyses in Methods section (Page 13, Line 8-9). Additionally, in the results section of our manuscript, we refrained to mention “significantly” when the statistical analysis did not show P < 0.05. 

8. In the Methods section, please provide the specific sequences of the primers used in the PCR analysis conducted in your study.

 Thank you so much for your important suggestion. We added some explanation about real-time RT-PCR in the present study (Page 9, Line 14-17). Actually, we adopted the TaqMan assay system, and all primers used in the analysis were purchased form Applied Biosystem. In contrast to the analysis using SYBR Green master-mix, specific sequences of the primers derived from Applied Biosystem were not officially presented. Thereby, we presented the specific assay ID for each primer instead of specific sequence (S1 Table). 

9. PLOS ONE now requires that authors provide the original uncropped and unadjusted images underlying all blot or gel results reported in a submission’s figures or Supporting Information files. This policy and the journal’s other requirements for blot/gel reporting and figure preparation are described in detail at https://journals.plos.org/plosone/s/figures#loc-blot-and-gel-reporting-requirements and https://journals.plos.org/plosone/s/figures#loc-preparing-figures-from-image-files.

When you submit your revised manuscript, please ensure that your figures adhere fully to these guidelines and provide the original underlying images for all blot or gel data reported in your submission. 

 I completely understood the journal’s policy and requirements for blot/gel reporting. All blot/gel image data shown in the present study were included in Supporting Information (S2 Raw images of all original blot and gel images), which was described in the cover letter.

10. Thank you for stating the following in your Competing Interests section: 

[No conflicts of interest are declared by the authors.]. Please complete your Competing Interests on the online submission form to state any Competing Interests. 

If you have no competing interests, please state "The authors have declared that no competing interests exist.", as detailed online in our guide for authors at http://journals.plos.org/plosone/s/submit-now This information should be included in your cover letter; we will change the online submission form on your behalf.

 As you indicated, we modified the statement of Competing Interests section as following; The authors have declared that no competing interests exist. Also, this information was included in the cover letter. 

Reviewer 1

Major comments

In the paper entitled “Reno-protective effect of IL-34 inhibition on cisplatin-induced nephrotoxicity in mice”, authors described the role and the modulation of IL-34 and its receptors during damage in an in vitro and in vivo model of renal damage, caused by cisplatin nephrotoxicity. Authors dissected the effect of the blocking of IL-34 using a blocking antibody on TEC and macrophages in vitro and in vivo. They demonstrated that in vivo, the use of anti-IL-34 ab reduced the expression of IL-34 itself and its receptors and displayed a beneficial effect on renal recovery after cisplatin damage. In particular, the treatment generated a reduction of tubulointerstitial injury, suppressed macrophage infiltration, and reduced apoptotic cell. In vitro authors demonstrated that cisplatin treatment on TEC induced the expression of IL-34, that in turn promoted the progression of damage. The conclusion is that IL-34 may be a potential therapeutic target, ant its inhibition may display renoprotective effects. The paper is well designed and well explained and clear. I would suggest some minor changes to improve the manuscript.

Major points: 

1. The “abstract” is too long, I suggest being more concise. 

 We appreciate your suggestion. We made effort to simplify the abstract as much as possible. In particular, we did delete the result about colony stimulating factor-1 (CSF-1) in the revised abstract since the explanation for CSF-1 was sufficiently mentioned in the manuscript body. Consequently, we reduced the total volume of abstract from 300 words to 267 words (Page 2, Line 1-Page 3, Line 11). 

2. In the introduction section the part describing macrophages (M1 and M2) should be summarized. 

 As you pointed out, description of the part regarding macrophages (M1 vs.M2) seemed to be redundant. In addition, some phrases or explanations were duplicated in the discussion section. Therefore, we sought to edit and simplify the part of introduction as much as possible (Page 4, Line 19-Page 5, Line 9). Consequently, we reduce the volume of that part from 232 words to 188 words. 

3. In the discussion section, I would suggest to be more discursive and add sentence before the point list.

 As you suggested, we added a sentence before the point list in the discussion section (Page 27, Line 1-3). Furthermore, description in the discussion section might be tedious on the whole, therefore we made effort to remove redundant text and tried to make the text easier to read so as not to provide wrong message or interpretation. Finally, we reduced the total; volume of discussion section from 1870 words to 1677 word.

Minor point:

1. Page 14, line 20, please change “dedicated” with specific

 Thank you so much for your suggestion, and “dedicated” used in that part was meant as “specific” or “recommended”. However, such explanation using “dedicated” might led readers to confuse. Moreover, “dedicated medium for MRPTEpiC was described as Epithelial Cell Medium-animal (EpiCM-a, #4131, ScienCell) in the Method section (Page 12, Line 2-3). Thus, we deleted its description from the manuscript (Page 15, Line 14). We consider that most readers could grasp the meaning even though we deleted the description of “dedicated”. 

We feel that we have addressed all of the concerns of the referees and that the manuscript has been significantly improved. We hope that you find the revised version of our manuscript suitable for publication in PLOS ONE.

We look forward to receiving your response.

Sincerely,

Yukihiro Wada

Division of Nephrology, Department of Medicine

Showa University School of Medicine

Tokyo, Japan

Address: 1-5-8 Hatanodai, Shinagawa-ku, Tokyo 142-8666, Japan

Tel: +81-3-3784-8533

E-mail address: yukihiro@med.showa-u.ac.jp

---

## [Decision Letter · Decision Letter 1]

29 Dec 2020

Reno-protective effect of IL-34 inhibition on cisplatin-induced nephrotoxicity in mice

PONE-D-20-32498R1

Dear Dr. Yukihiro Wada,

We’re pleased to inform you that your manuscript has been judged scientifically suitable for publication and will be formally accepted for publication once it meets all outstanding technical requirements.

Kind regards,

Benedetta Bussolati, MD, PhD

Academic Editor

PLOS ONE

Additional Editor Comments (optional):

Reviewers' comments:

Reviewer's Responses to Questions

**Comments to the Author**

1. If the authors have adequately addressed your comments raised in a previous round of review and you feel that this manuscript is now acceptable for publication, you may indicate that here to bypass the “Comments to the Author” section, enter your conflict of interest statement in the “Confidential to Editor” section, and submit your "Accept" recommendation.

Reviewer #1: All comments have been addressed

2. Is the manuscript technically sound, and do the data support the conclusions?

Reviewer #1: Yes

3. Has the statistical analysis been performed appropriately and rigorously? 

Reviewer #1: Yes

4. Have the authors made all data underlying the findings in their manuscript fully available?

Reviewer #1: Yes

5. Is the manuscript presented in an intelligible fashion and written in standard English?

Reviewer #1: Yes

6. Review Comments to the Author

Reviewer #1: The authors addressed all of the points; the manuscript in the present form is accepted for pubblication. It has been significantly improved.

7. PLOS authors have the option to publish the peer review history of their article (what does this mean?). If published, this will include your full peer review and any attached files.

Reviewer #1: No

---

## [Editor Report · Acceptance letter]

2 Jan 2021

PONE-D-20-32498R1 

Reno-protective effect of IL-34 inhibition on cisplatin-induced nephrotoxicity in mice 

Dear Dr. Wada:

I'm pleased to inform you that your manuscript has been deemed suitable for publication in PLOS ONE. Congratulations! Your manuscript is now with our production department. 

Kind regards, 

on behalf of

Prof. Benedetta Bussolati 

Academic Editor

PLOS ONE